# The Dynamic Shift of Bacterial Communities in Hybrid Anaerobic Baffled Reactor (ABR)—Aerobic Granules Process for Berberine Pharmaceutical Wastewater Treatment

**Yan Wang** [1,2], **Yongqiang Liu** [3], **Juan Li** [1,2], **Ruirui Ma** [1,2], **Ping Zeng** [1,2,*], **Choon Aun Ng** [4] **and Fenghua Liu** [5]

1   State Key Laboratory of Environmental Criteria and Risk Assessment, Chinese Research Academy of Environmental Sciences, Beijing 100012, China
2   Department of Urban Water Environmental Research, Chinese Research Academy of Environmental Sciences, Beijing 100012, China
3   Faculty of Engineering and Physical Sciences, University of Southampton, Southampton SO17 1BJ, UK
4   Department of Environmental Engineering, Faculty of Engineering and Green Technology, Universiti Tunku Abdul Rahman, Kampar 31900, Perak, Malaysia
5   CECEP Engineering Technology Research Institute Co., Ltd., Beijing 100082, China
*   Correspondence: zengping@craes.org.cn

**Abstract:** Because of its anticancer, anti-inflammatory, and antibiotic properties, berberine has been used extensively in medication. The extensive production of berberine results in the generation of wastewater containing concentrated residual berberine. However, to date, limited related studies on the biological treatment of berberine wastewaters have been carried out. A lab-scale anaerobic baffled reactor (ABR)–aerobic granular sludge (AGS) process was developed for berberine removal from synthetic wastewater. The system showed effective removal of the berberine. In order to better understand the roles of the bacterial community, the ABR–aerobic granular sludge system was operated in the state with the highest BBR removal rate in this study. The bacterial community dynamics were studied using the 16S rDNA clone library. The results showed that the hybrid ABR-AGS process achieved 92.2% and 94.8% overall removals of berberine and COD, respectively. *Bacterium* was dominant species in ABR, while the *CFB group bacteria* and *Betaproteobacteria* were dominant species in AGS process. The *uncultured bacterium clone B135*, *Bacillus endophyticus strain a125*, *uncultured bacterium mle1-42*, *uncultured bacterium clone OP10D15*, and *uncultured bacterium clone B21.29F54* in ABR, and *uncultured bacterium clone F54*, *uncultured bacterium clone ZBAF1-105*, *uncultured bacterium clone SS-9*, and *uncultured bacterium clone B13* in AGS process were identified as functional species in the biodegradation of berberine and/or its metabolites. Both anaerobic and aerobic bacterial communities could adapt appropriately to different berberine selection pressures because the functional species' identical functions ensured comparable pollutant removal performances. The information provided in this study may help with future research in gaining a better understanding of berberine biodegradation.

**Keywords:** 16S rDNA clone library; bacterial community structure; berberine wastewater; anaerobic baffled reactor (ABR); aerobic granular reactor

## 1. Introduction

Berberine (5,6-dihydro-9,10-dimethoxybenzo [g]-1,3-benzodioxolo [5,6-$\alpha$]) quinolizinium ($C_{20}H_{18}NO_4$, abbreviated BBR) is an isoquinoline quaternary alkaloid. BBR can be extracted from herbal plants, or chemically synthesized, and used as a natural antibiotic against a variety of bacteria [1]. BBR's application has been expanded to antitumor, anti-oxidation, anti-disease, anti-Alzheimer's, and anti-hyperglycemic due to its anticancer, anti-inflammatory, and antibiotic properties [2,3], resulting in a sharp increase in the demand for BBR. Meanwhile, the widespread production and use of BBR has resulted in the discharge of large amounts of BBR-containing wastewater into the environment (thousands

of mg/L). Berberine's IC50, minimum inhibitory concentration (MIC), and minimum micro-bicidal concentration (MMC) values against resistant *Pseudomonas aeruginosa* and *Escherichia coli* were found to be 99.2, 240, and 250 mg/L, and 87.0, 469, and 500 mg/L, respectively [4]. Thus, a significant threat is posed to ecosystems due to its significant inhibitory effects on biological activities [5]. As a result, prior to its discharge into the environment, BBR in wastewater, particularly that from pharmaceutical processes, must be treated.

Physical, chemical, and biological processes are commonly used to treat BBR-containing wastewater [5–7]. Due to their high cost and risk of producing new pollutants, physical and chemical treatments of BBR-containing wastewater have limited application [8,9]. As a result, biological treatment is preferred, due to its lower cost and its potential for complete mineralization. The application of a combined anaerobic–aerobic treatment system is recommended for the treatment of wastewater containing antibiotics and pharmaceutical effluents in the guidelines on the available techniques of pollution prevention and the control of the pharmaceutical industry active pharmaceutical ingredient (fermentation, chemical synthesis, and extraction) and preparation categories [10].

Anaerobic baffled reactor (ABR) showed a better resistance to toxic compounds, which is attractive for the pharmaceutical wastewater treatment. For ABR, every chamber served as a UASB, shielding the vulnerable microorganisms from the toxic substrate [11,12]. Meanwhile, the push-flow pattern is similar to the overall reactor flow pattern. Because of its unique structure, it can process toxic substances and hard-to-degrade inhibitors with a better buffering adaptability. As for the aerobic processes, aerobic granular sludge was found to be a promising alternative in eliminating the recalcitrant and toxic antibiotics [13–15]. Thus, a hybrid ABR–AGS system was setup in order to degrade BBR-containing wastewater. The ABR–AGS system showed a higher endurance with a higher influent COD concentration compared to an up-flow anaerobic sludge blanket (UASB)–membrane bioreactor (MBR) system [16,17]. However, in most industrial pharmaceutical production plants, the anaerobic process showed a low removal efficiency effected by the influent toxic compounds. It is necessary to make clear the roles of the microorganisms during the BBR degradation process in order to understand the mechanisms of BBR degradation.

In this study, the ABR–aerobic granular sludge system was operated in the state with the highest BBR removal rate. The bacterial community dynamics were studied in order to further improve the system degradation efficiency by using the 16S rDNA clone library, which is still widely used for microbial community studies in wastewater treatment processes [18,19]. Theoretical guidance is provided in order to further improve the treatment capacity and the stability of aerobic granular sludge. The goal of this study was to provide some basic information on the bacterial community composition in the biological berberine treatment process. It is anticipated to serve as a resource for further pilot-scale or industrial wastewater treatment.

## 2. Materials and Methods

### 2.1. Hybrid ABR–AGS System

The ABR (Figure S1) was made in the shape of a rectangle, with 610 mm length, 300 mm width, and 430 mm height, which provided the effective volume of 30 L [20–22]. The reactor consisted of 4 chambers. The widths of the upper flow chamber and the lower flow chamber were 90 mm and 30 mm, respectively [23,24]. The angle of the baffle plate set at 45° to get a higher lower flow speed, which could push the sludge in the bottom of reactor up to the floating level so that the sludge and liquid could be mixed completely. The sampling ports were set at the top and bottom of each chamber and were used to take supernatant and sludge, respectively. A gas collection port was arranged at the top of each chamber. The temperature of the reactor was kept at $32 \pm 1$ °C by binding the reactor walls with a tubular heater.

A sequencing batch reactor (SBR) was a cylinder with 600 mm internal diameter and 1000 mm height, with H/D of 16:1, providing an effective volume of 2.8 L (Figure S1).

Reactors were operated periodically at 4 h as one cycle. Every cycle included influent time of 4 min, aeration time of 225 min, settling time of 5 min, and effluent time of 5 min.

The effluent was discharged from a port with 50 cm distance from the bottom of the reactor, thus, the volumetric exchange ratio was 50%. A total of 4.0 L min$^{-1}$ of aeration was provided by an air dispenser located at the bottom of the reactor. The hydraulic retention time (HRT) was 8 h.

No excess sludge was discharged from the hybrid ABR–AGS system.

### 2.2. Inoculum

Inoculums, with an initial concentration of 13,570 mg MLSS/L for the operation of ABR, were obtained from a chemical synthetic pharmacy company's wastewater treatment plant (hydrolysis/acidification tank) in Shenyang, China. As shown in Table S1, the inoculum of SBR, with an initial concentration of 2350 mg MLSS/L, was obtained from the aeration tank of the same pharmaceutical wastewater treatment plant.

The hybrid ABR–AGS system's operation conditions are summarized in Table S2.

### 2.3. Medium

The hybrid ABR–AGS system was fed with synthetic wastewater with the components of glucose and the industrial berberine wastewater, providing COD of 4253 ± 102 mg/L and berberine of 121.6 ± 2.4 mg/L. The effluent from ABR was provided to SBR as an influent. The high-concentration berberine mother liquid was discharged from the separation process during berberine production as the main component of the industrial berberine wastewater. The concentration of the berberine mother liquid was quite high, with COD of 4166 ± 102 mg/L and berberine of 900 ± 100 mg/L [25]. The compositions of the influent wastewater are summarized in Table S3.

### 2.4. Analytical Methods

The samples of the ABR and SBR influent and effluent were taken every day. Standard methods were used to determine the COD and $NH_4^+$-N contents of the wastewater samples. The limit of quantitation (LOD) for COD and $NH_4^+$-N were 5.0 mg/L and 0.025 mg/L, respectively [26]. High performance liquid chromatography (HPLC) was used to determine the concentration of berberine (Agilent 1100, Santa Clara, CA, USA) at 345 nm (LOQ is 0.05 mg/L), with a 0.05 M $KH_2PO_4$/Acetonitrile (30:70 $v/v$) solution as the mobile phase. The flow rate was 1.0 mL/min. A 0.4 μm polytetrafluoroethene microfiltration membrane was used to filter 20 μL of the wastewater sample, which was injected with an auto-sampler into HPLC for analysis. A column of Agilent HB-C8 (150 mm 4.6 mm, 5 μm) was used to separate analytes at 30 °C [25].

### 2.5. Sampling and DNA Extraction

The clone libraries were constructed using biomass samples that were collected at different operation times [27]. The floc sludge levels in ABR and SBR were sampled separately, and the aerobic granules were dispersed using sonication and suspended in sterile water. After centrifuging all of the samples for 5 min at $1000 \times g$, the supernatant was decanted and the pellet was re-suspended in Tris-EDTA buffer (10.0 mM Tris-HCl, 1.0 mM EDTA, pH 8.0). After resuspension, all samples were immediately frozen and stored at −80 °C until DNA extraction.

According to the manufacturer's instructions, DNA was extracted from the samples using a QIAamp DNA Mini Kit (Qiagen, Valencia, CA, USA). The DNA that was extracted in triplicate for each sample was mixed to create the templates used for PCR amplification in order to reduce variations in DNA extraction.

### 2.6. PCR Amplification and 16S rDNA Cloning

The universal primers 27F and 1492R were used to amplify the 16S rDNA genes from the DNA extracts [11]. A DNA thermocycler (Bio-Rad, Richmond, CA, USA) was used to

amplify PCR in a total volume of 50 L in 200 L tubes. Every sample for PCR contained 40 ng of template DNA, 200 μM of each deoxynucleoside triphosphate, 0.5 μmol of each primer, 1.25 U of Taq polymerase (Pro- mega, Madison, WI, USA), $1 \times$ PCR buffer, and 2 mM MgCl$_2$. The temperature cycling conditions were as follows: pre-incubation at 95 °C for 2 min, 25 cycles of 95 °C for 1 min, 62 °C for 1.5 min, and 72 °C for 1 min, followed by 10 min at 72 °C.

Following the manufacturer's instructions, a Qiaquick PCR cleanup kit (Qiagen, Valencia, CA, USA) was used to purify the PCR products, which was ligated into a PCR 2.1-TOPO vector, then transformed into TOP 10 *E. coli* competent cells (Invitrogen, Carlsbad, CA, USA). For each sample, approximately 100 clones were randomly selected for analysis using ampicillin and x-gal.PCR amplification. The PCR amplification with the primer pair M13 was used to identify positive clones, which used the same program as for 16S rDNA amplification. Sequencing was performed using an ABI 3730 automated sequencer for all positive clones (Invitrogen, Carlsbad, CA, USA).

## 3. Results

### 3.1. The Performance of Hybrid ABR–AGS System

The hybrid ABR–AGS system was run for 125 days with berberine concentrations of 121.62.4 mg/L as the influent. The berberine and COD removal rates were high and stable overall. Table 1 summarizes the system's treatment performance. A significant proportion of the berberine (57.0 $\pm$ 0.2%) and COD (71.9 $\pm$ 1.0%) removal rates was achieved in ABR. The SBR containing aerobic granules removed COD and berberine at rates of about 81.8 $\pm$ 0.3% and 81.6 $\pm$ 0.5%, respectively, resulting in effluent COD and berberine levels of less than 219.7 mg/L and 9.5 mg/L, respectively. These findings suggest that the hybrid ABR–AGS system was effective at both berberine reduction and COD removal. Furthermore, these findings indicate that a functional stable bacterial community for berberine degradation had already been established. The biomass samples in the ABR and SBR reactors were subjected to 16S rDNA clone library analysis in order to better understand the bacterial community composition and to identify the key functional bacterial species/groups in the berberine biodegradation process.

**Table 1.** The results of the berberine wastewater treatment in the hybrid ABR–AGS system.

| | Influent (mg/L) | ABR Effluent (mg/L) | AGS Effluent (mg/L) | ABR Removal Efficiency (%) | AGS Removal Efficiency (%) | Overall Removal Efficiency (%) |
|---|---|---|---|---|---|---|
| Berberine | 121.6 $\pm$ 2.4 | 52.3 $\pm$ 1.3 | 9.5 $\pm$ 0.4 | 57.0 $\pm$ 0.2 | 81.8 $\pm$ 0.3 | 92.2 $\pm$ 0.7 |
| COD | 4253 $\pm$ 104 | 1193.1 $\pm$ 46 | 219.7 $\pm$ 2.4 | 71.9 $\pm$ 1.0 | 81.6 $\pm$ 0.5 | 94.8 $\pm$ 1.3 |

### 3.2. Differences in the Structure of Total Bacterial Communities in ABR Reactor Chambers

At the steady operation condition, with the feeding influent berberine concentration of 120 mg/L, the 16S rDNA clone libraries of the total bacteria were produced from the sludge samples in each chamber of the ABR reactor. After the comparison study, the total bacterial clone libraries in each chamber of the ABR reactor were 31, 32, 35, and 38 operational taxonomic units (OTUs) during stable operation. The results were shown in Figure 1.

Under the stable operating conditions, there were 31 OTUs of total bacteria in the A1 chamber of the ABR reactor, with a coverage of 84%, and a Shannon–Wiener diversity index of 2.84. These clones had a 99% maximum similarity and an 85% minimum similarity to the known bacteria in the Gen-bank. According to an analysis of the 31 OTU sequence data, the 31 OTUs belonged to six different taxa within the bacterial domain. The *uncultured bacterium partial 16S rRNA gene* from *clone 053B03_B_DI_P58* was the most dominant group, accounting for 24% of the total bacterial flora, and its clone was 98% similar to the known bacteria in the Gen-bank; The *uncultured Bacteroides* sp. *Bacteroides* sp. *clone J3* (*Bacteroides* sp.) and the *uncultured bacterium 054B06_B_DI_P58*, were the second abundant group, accounting for 11% and 9% of the total bacterial community, respectively. The

*uncultured bacterium clone B135* and *Bacillus endophyticus strain a125* with antibiotic resistance propertiesaccounted for 6% and 2% of the total bacterial flora, respectively; Each of the *uncultured bacterium clone DC75* and the *uncultured Clostridia bacterium clone L24* (*Clostridium perfringens*) accounted for 3% of the total bacterial flora and both of them were hydrolytic acidifying functional colonies.

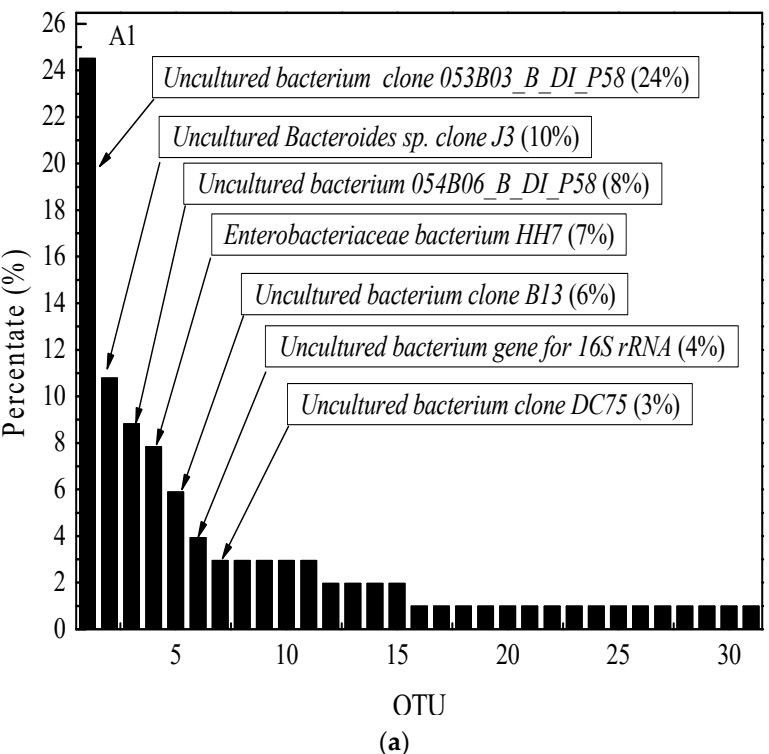

(**a**)

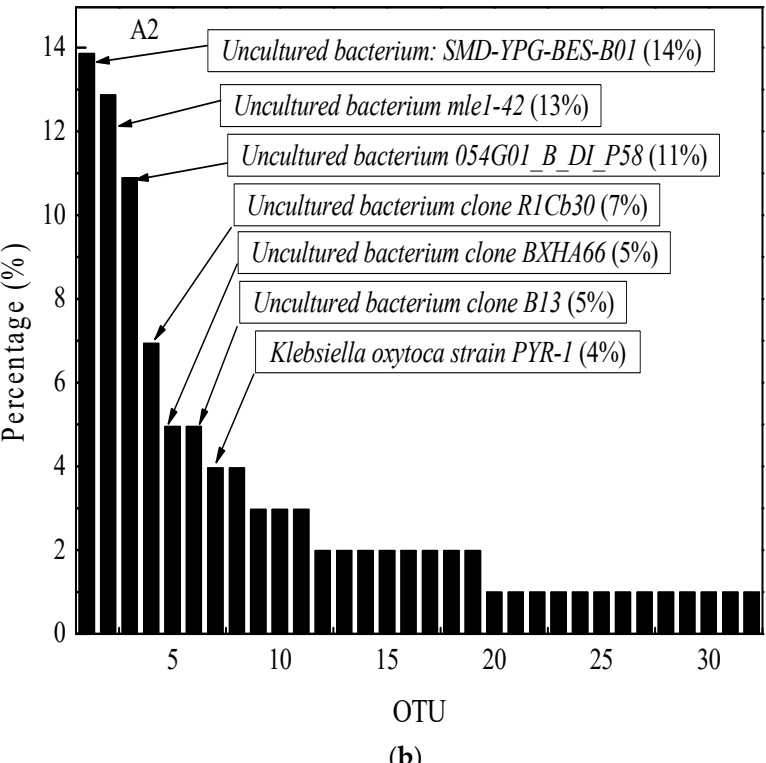

(**b**)

**Figure 1.** *Cont.*

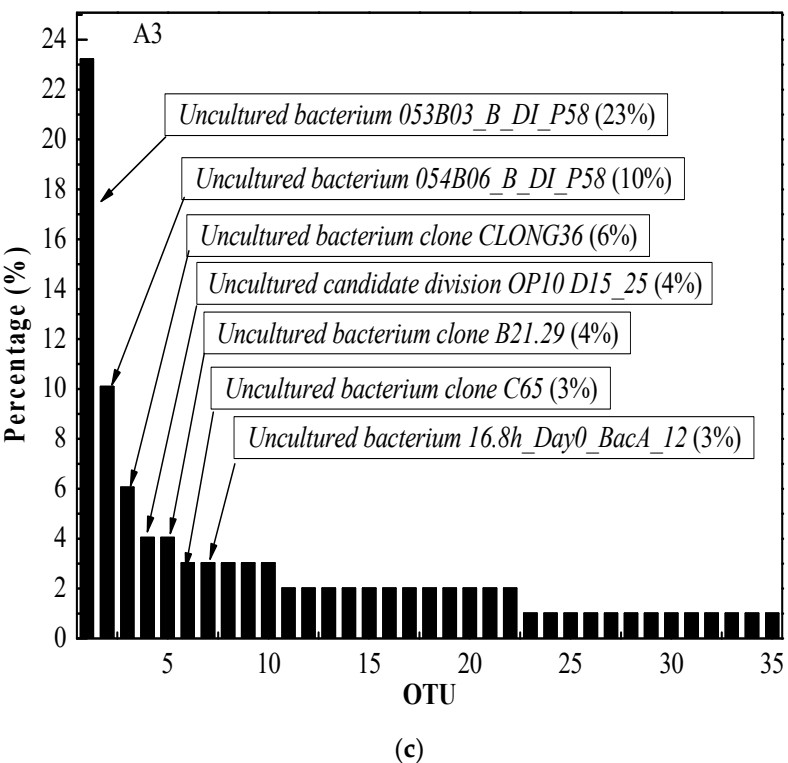

(**c**)

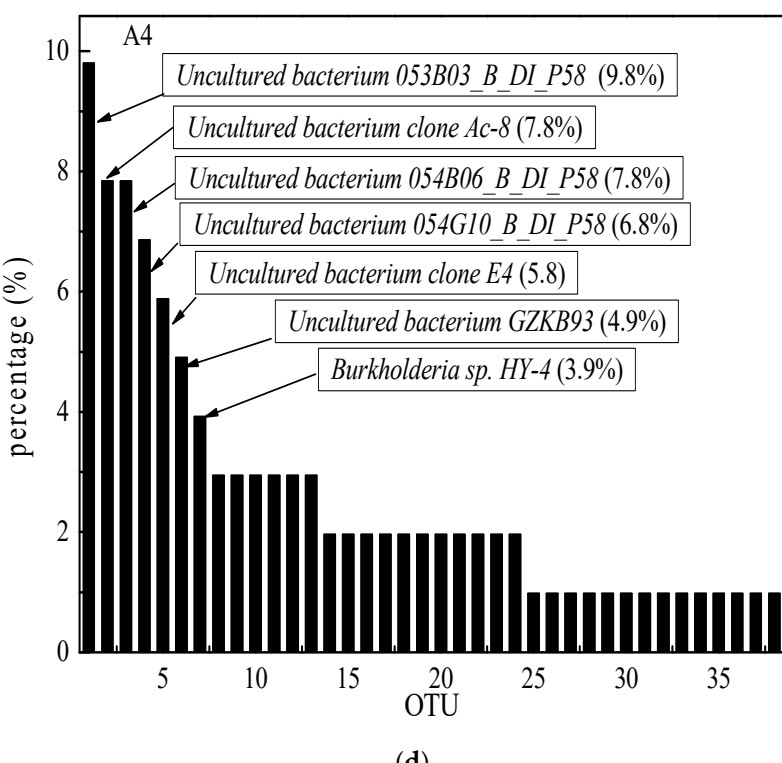

(**d**)

**Figure 1.** The total bacterial community in compartments of ABR (**a**) chamber A1, (**b**) chamber A2, (**c**) chamber A3, and (**d**) chamber A4.

For chamber A2, the OTU number of total bacteria was 32, with a coverage of 87%, a Shannon–Wiener diversity index of 3.04, and a minimum similarity of 86% to the known sequences, belonging to five taxa. The top three dominant groups were the *uncultured bacterium SMD-YPG-BES-B01*, the *uncultured bacterium mle1-42*, and the *uncultured bacterium 054G01_B_DI_P58*, with 14%, 13%, and 11% of the total bacterial community of the system, respectively. The *uncultured bacterium mle1-42* was the dominant colony for the degradation of the pharmaceutical wastewater pollutants. The bacterial communities with antibiotic resistance increased to 16% of the total bacterial community. The *uncultured bacterium gene for 16S rRNA* accounted for 1% of the total bacterial community and was a volatile fatty-acid-producing community.

For chamber A3, the OTU number of total bacteria was 35, with 86% coverage, and the Shannon–Wiener diversity index was 3.07, with a minimum similarity of 90% to the known sequence comparisons, belonging to five taxa. Among them, the *uncultured bacterium 053B03_B_DI_P58* was the most dominant group, accounting for 23% of the total bacteria, and was an anaerobic archaeal community. The next dominant colonies were the *uncultured bacterium 054B06_B_DI_P58* and the *uncultured bacterium clone CLONG36*, accounting for 10% and 6% of the total bacterial community, respectively, with the *uncultured bacterium clone CLONG36* belonging to the anaerobic granular sludge community. The bacterial communities with antibiotic resistance accounted for 7% of the total bacterial community. *Bacterium K-4b6*, which are acidophilic alkane-producing bacteria, accounted for 1% of the bacterial community.

For chamber A4, the OTU number of total bacteria was 38, with 86% coverage, and the Shannon–Wiener diversity score was 3.35, with a minimum similarity of 83% to the known sequence matches, belonging to four taxa. The *uncultured bacterium 053B03 B DI P58* was the most prevalent, accounting for 9.8% of the total microorganisms. *Acidophilic alkane-producing bacteria* made up 3% of the total bacterial population.

A phylogenetic tree was constructed for the bacteria in each chamber of the ABR reactor, and the results are shown in Figure 2.

(1)    The total bacteria in chamber A1 of the ABR reactor:

In A1, 80 clones were retrieved and were grouped into 23 operational taxonomic units (OTUs). *Bacteria* was the dominant species, accounting for 68% of the total bacterial community; *CFB group bacteria* and *firmicutes* were the next two dominant species, each accounting for 11% of the total bacterial community;

(2)    The total bacteria in the A2 chamber of the ABR reactor:

In A2, 87 clones were retrieved and were grouped into 25 OTUs. *Bacteria*, as the dominant species, accounted for 86% of the total bacterial community;

(3)    The total bacteria in chamber A3 of the ABR reactor:

In A3, 91 clones were retrieved and were grouped into 30 OTUs. *Bacteria* were the dominant species, accounting for 92% of the total bacterial community;

(4)    The total bacteria in chamber A4 of the ABR reactor:

In A4, 90 clones were retrieved and were grouped into 33 OTUs. *Bacteria* were the dominant species, accounting for 92% of the total bacterial community. *Characteria* served as the dominant species, accounting for 88% of the total bacterial community.

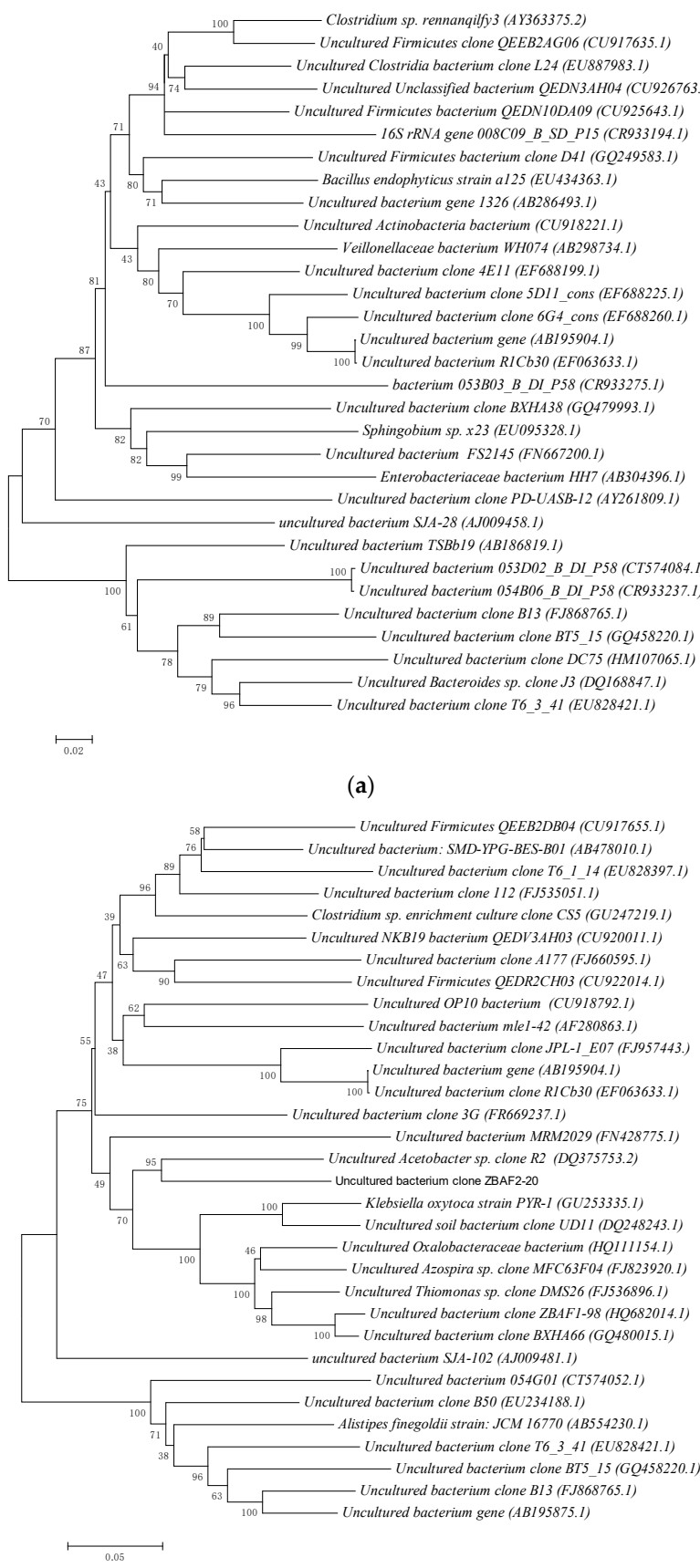

(**a**)

(**b**)

**Figure 2.** *Cont.*

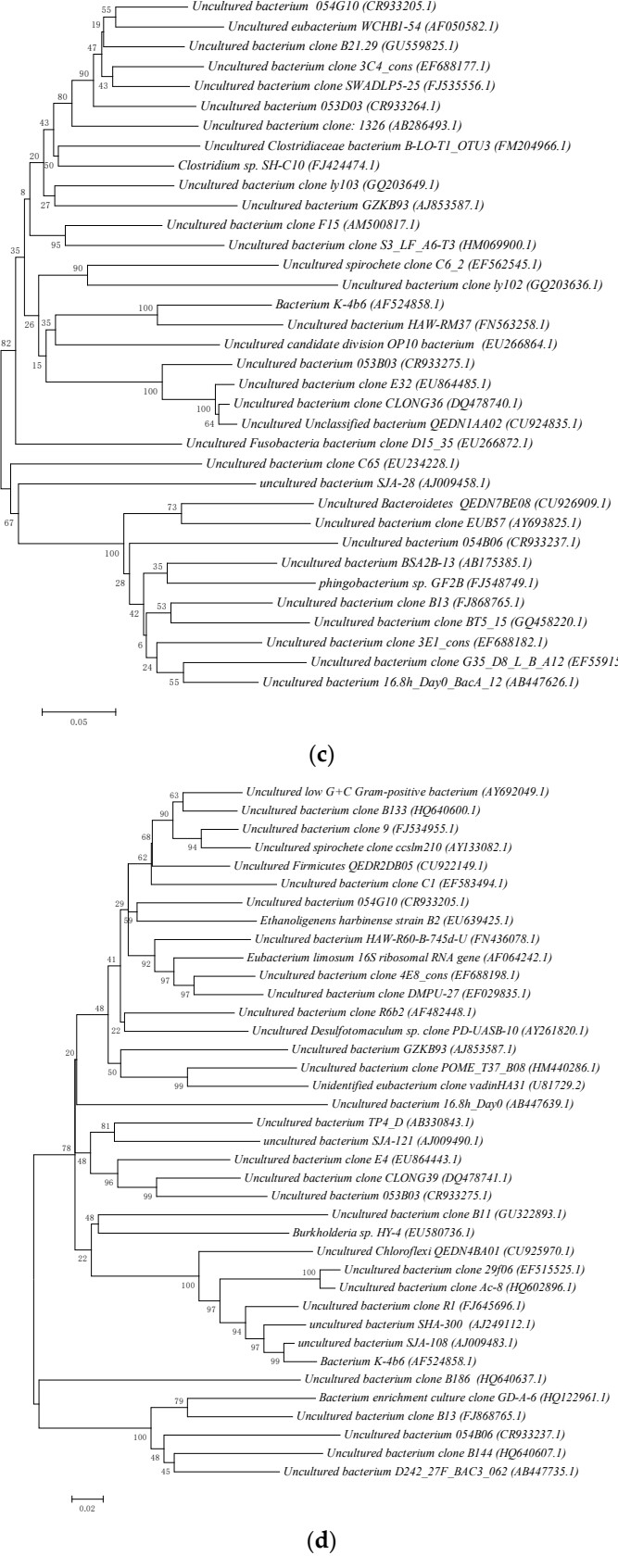

(**c**)

(**d**)

**Figure 2.** Phylogenetic tree for the total bacterial community in ABR chambers (**a**) chamber A1, (**b**) chamber A2, (**c**) chamber A3, and (**d**) chamber A4.

*3.3. Differences in the R3 Reactor Sludge Granulation Process's Overall Bacterial Community Structure*

The total bacterial 16S rDNA clone libraries were constructed for the sludge samples at different operation times of the aerobic granulation process. Each library contained about 100 clones (with a sequence length of about 1400 bp), and the sequencing results were compared by BLAST in the Gen-bank. The total bacterial clone library in the aerobic granular sludge (AGS) reactor after 15, 40, and 80 reactor starts was 38, 34, and 35 OTUs, respectively. The results are shown in Figure 3.

When the AGS reactor started up, the aerobic granular sludge appeared after 15 days of operation. The OTU number of total bacteria in the reactor was 38, with 77% coverage, and the Shannon–Wiener diversity index was 3.01. The maximum similarity between these clones and the known bacteria in the Gen-bank was 100%, while the minimum similarity was 90%. Among them, OTU1 was the uncultured bacterial clone. In addition, the most dominant species was the *Uncultured bacterium clone A156*, which accounted for 28% of the total bacterial community, and its clones were 93% similar to the known bacteria in the Gen-bank. It was followed by the *Uncultured bacterium clone AS-19* and *Hydrogenophaga sp. EMB 7* as the dominant species, which accounted for 8% and 6% of the total bacterial community, respectively. The *uncultured bacterium clone F54*, *Uncultured bacterium clone ZBAF1-105*, and the other antibiotic-resistant bacterial communities accounted for 11% of the total bacterial community; the *Uncultured bacterium clone M01* accounted for 3% of the total bacterial community by enhancing the sludge settling ability, and the *Bacterium clone M0111 48*, which represented for 3% of the overall bacterial community, helped with the generation of the aerobic granular sludge.

After 40 days of operation, the aerobic granular sludge was the dominant form in AGS reactor. The OTU number of the total bacteria in the AGS was found to be 34, with 89% coverage, a Shannon–Wiener diversity index of 3.21, and a minimum similarity of 90% to the known sequence comparisons, belonging to six taxa. The *uncultured Bacteroidetes bacterium clone Skagenf54 bacteria* was the most dominant species, belonging to *CFB group bacteria*, accounting for 19% of the entire bacterial community. The second most dominant species was the *Uncultured bacterium clone SS-9*, accounting for 4% of the overall bacterial community, and belonging to the pyridine, quinoline, and derivative degradation group. The dominant species for the granulation in the aerobic granular sludge was the *Uncultured bacterium clone EUB72-2*, which consisted of uncultured clones with the same percentage as the *Uncultured bacterium clone SS-9*. At the same time, the microbial community of the *uncultured bacterium clone 77*, which was fitted for the nutrients cyclical changing operation mode, was generated, accounting for 4% of the total bacterial community. The antibiotic-resistant bacterial populations then accounted for 16% of the entire bacterial community.

After 80 days' operation, the AGS reactor was filled with mature aerobic granular sludge. The OTU number of total bacteria in the AGS was found to be 35, with 82% coverage, and the Shannon–Wiener diversity index was 3.11, with a minimum similarity of 95% to the known sequences from eight taxa. *Comamonas* sp. *PP3-1*, which is a *Betaproteobacteria* of the phylum Amoeba, was the most abundant group, accounting for 20% of the total bacteria. The following dominant species were *Comamonas* sp. *XJ-L67*, the *uncultured bacterium clone A_SBR_1*, and the *uncultured bacterium clone B13*, which accounted for 8%, 7%, and 5% of the total bacterial community, respectively. As one of these, the *uncultured bacterium clone B13* made up the majority of the bacterial communities with antibiotic wastewater treatment capability. The antibiotic-resistant bacterial populations made up 15% of the entire bacterial community.

A phylogenetic tree was constructed for the bacteria in the AGS reactor at different operation times, and the results are shown in Figure 4.

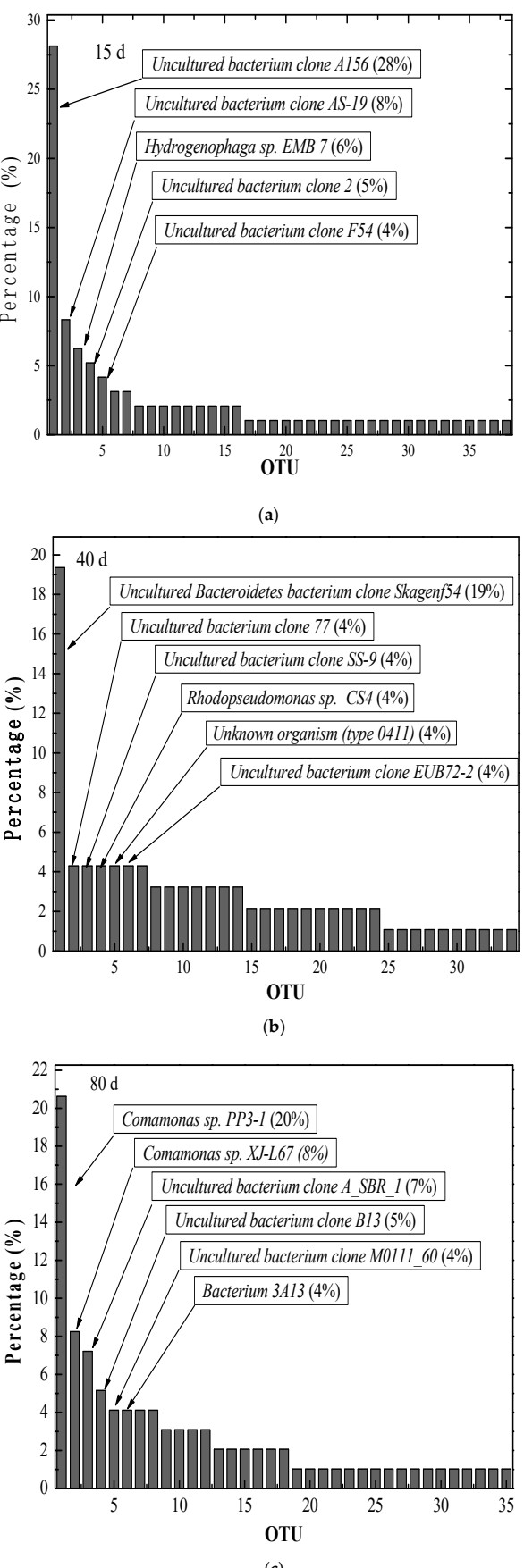

**Figure 3.** The total bacterial community in AGS at different stages (**a**) 15 d, (**b**) 40 d, and (**c**) 80 d.

(1)     The AGS reactor with 15 days of operation:

In the early stage of aerobic granular sludge formation, the *bacteria* and the *CFB group bacteria* dominated the reactor's bacterial ecology, accounting for 84% and 6% of the overall bacterial community, respectively;

(2)     The AGS reactor with 40 days of operation:

The aerobic granular sludge was the dominant form after 40 days of AGS reactor startup. The *Bacteria* and the *CFB group bacteria* were the main bacterial species in the reactor, accounting for 60% and 19% of the total bacterial community, respectively. The *a-proteobacteria* were close following them, accounting for 13% of all bacteria;

(3)     The AGS reactor with 80 days of operation:

When the AGS reactor ran for 80 days, the *Bacteria* and the *Betaproteobacteria* dominated the bacterial community, accounting for 43% and 34% of the total bacterial community, respectively.

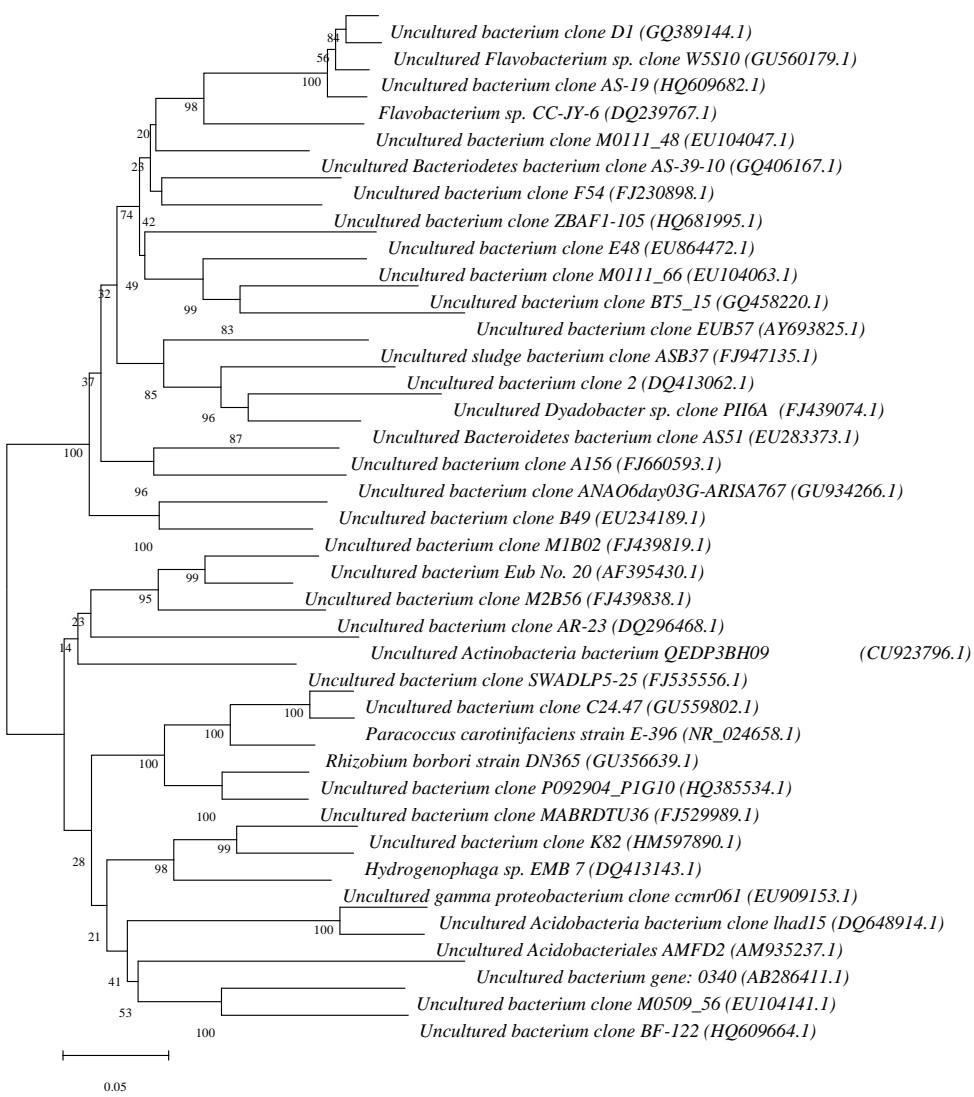

(**a**)

**Figure 4.** *Cont*.

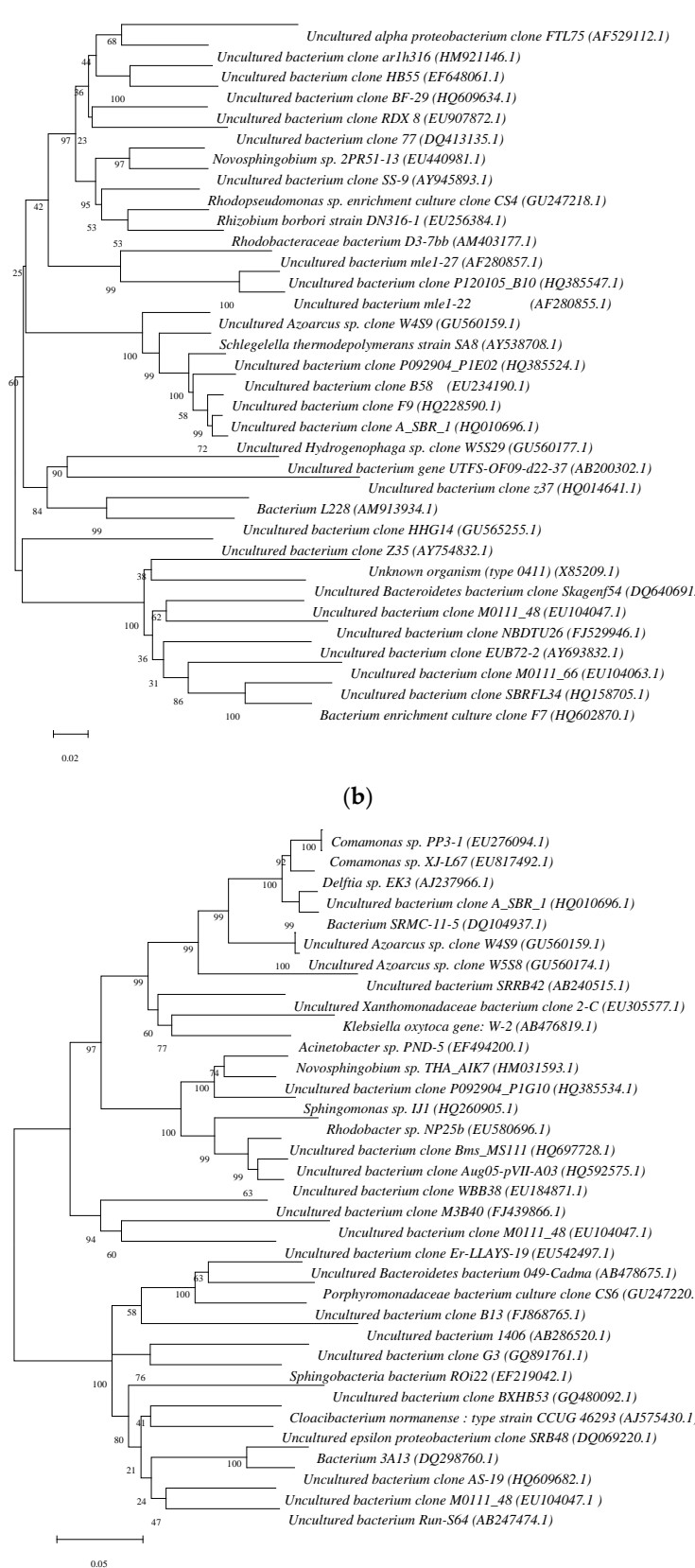

**Figure 4.** Phylogenetic tree for the total bacterial community in AGS at different stages (**a**) 15 d, (**b**) 40 d, and (**c**) 80 d.

## 4. Discussion

Berberine is an anti-inflammatory quaternary ammonium salt [2,3]. A high concentration of berberine is toxic to microbial organisms because it damages their cytoplasmic membranes and deactivates their enzymes. Berberine removal from synthetic wastewater was proposed using a hybrid ABR–AGS process. The promising results indicated that the hybrid ABR–AGS process is feasible and efficient in the degradation of berberine. The dynamic shift of bacterial communities in the ABR–AGS system is shown in Figure 5.

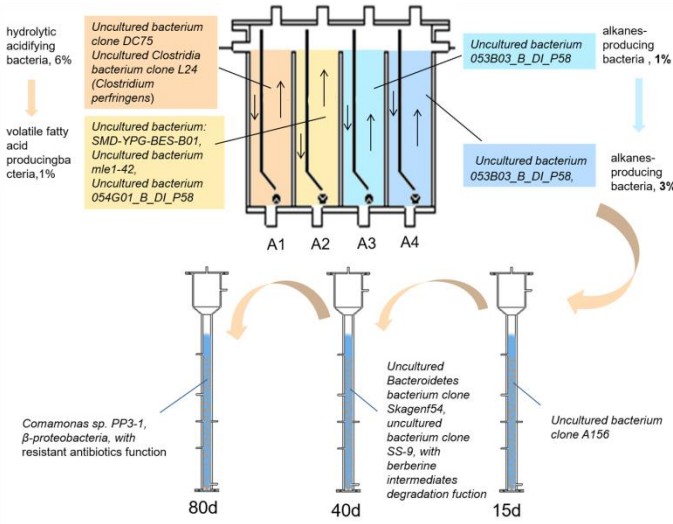

**Figure 5.** The schematic diagram of dynamic shift of bacterial communities in the hybrid ABR–AGS system.

When the influent berberine concentration was 120 mg/L, the dominant community in the ABR shifted as follows: *clone 053B03_B_DI_P58* (chamber A1) → *uncultured bacterium SMD-YPG-BES-B01, uncultured bacterium mle1-42, uncultured bacterium 054G01_B_DI_P58* (chamber A2) → *uncultured bacterium 053B03_B_DI_P58* (chamber A3) → *uncultured bacterium 053B03_B_DI_P58* (chamber A4). The results were consistent with the results that were provided by the phylogenetic tree that was constructed by the 16S rDNA clone library methods. The function of the chambers changed from hydrolytic acidification to methane production, with the influent flowing from A1 to A4. Thus, the colonies of the *uncultured bacterium clone DC75* and the *uncultured Clostridia bacterium clone L24* (*Clostridium perfringens*) with hydrolytic acidifying function occupied 6% of the total bacterial community in the chamber. *Clostridium* is a *Firmicutes* genus that has a hard cell wall and can produce endospores [28,29]. In addition, *Clostridium* was observed in a UASB reactor feed, with berberine as a carbon source [26]. There was a high proportion of Gram-positive low GC bacteria in the other anaerobic bioreactors as well [30]. Some *Clostridium* strains were found to be capable of using the methyl group of aromatic methyl ethers as a carbon source via O-demethylation [29]. It has also been reported that another stain can cleave aromatic rings [28]. Thus, there may be a functional species that is involved in the cleavage of the aromatic rings and/or the degradation of the methoxyl groups in berberine molecules.

In chamber A2, the *uncultured bacterium gene* of 16S rRNA with the function of volatile fatty acid production decreased to 1% of the total bacterial community. While *Bacteria*, as the dominant species, accounted for 86% of the total bacterial community.

In chamber A3, the acidophilic alkane-producing bacteria consisted of 1% of the bacterial community, indicating that the function of the microorganism changed from acid production to alkane production.

In chamber 4, the rate of alkane-producing bacteria in the total bacterial community increased to 3%. Since berberine is a drug with an antibiotic function, the rate of microorganisms with antibiotic resistance in the total bacterial community increased from 8% in chamber A1 to 16% in A2 and decreased to 7% in A3. Thus, most of the berberine was

degraded in chamber A1 and A2. The results were consistent with the result of berberine degradation in the ABR (Figure S2).

For the AGS operation, the aerobic granules appeared on day 15, dominated on day 40, and matured on day 80. The dominant community in the AGS shifted as follows: *uncultured bacterium clone A156 → uncultured Bacteroidetes bacterium clone Skagenf54 → Comamonas sp. PP3-1*. The rate of antibiotic-resistant microorganisms in the total bacterial community was found to change with operation time, increasing from 11% on day 15 to 16% on day 40, and remaining at 15% until day 80, indicating that the AGS's antibiotic resistance capability increased with the operation time. After day 40 of operation, an *uncultured bacterium clone SS-9* appeared, which was responsible for the degradation of pyridine, quinoline, and derivatives, perhaps because the degradation intermediates of berberine, pyridine, quinoline, and derivatives were aromatic compounds. The communities that were responsible for aerobic granules' formation were important. The *Uncultured bacterium clone M0111_48* accounted for 3% of the total bacterial community after 15 days of operation and the *Uncultured bacterium clone EUB72-2* accounted for 4% of the total bacterial community after 40 days of operation.

The *Proteobacteria* species was observed after 40 days of operation. The *α-Proteobacteria* appeared after 40 days of operation in the AGS reactor, while *β-Proteobacteria* appeared after 80 days of operation in the AGS reactor. According to Xia et al. (2012) and Zhang et. al. (2004), the bacterial strains that have antibacterial properties are important in wastewater treatment. [31,32]. Thus, *β-proteobacteria* may be the functional group to resistant antibiotics.

## 5. Conclusions

For the synthetic wastewater containing 120 mg/L berberine, the hybrid ABR–AGS process achieved 92.2% and 94.8% overall removals of berberine and COD, respectively. *Bacteria*, *CFB group bacteria*, and *Betaproteobacteria* dominated the AGS system, whereas the *Bacterium* dominated the ABR. The *Uncultured bacterium clone B135*, *Bacillus endophyticus strain a125*, *Uncultured bacterium mle1-42*, *Uncultured bacterium clone OP10D15*, and *Uncultured bacterium clone B21.29F54* in the ABR, and the *Uncultured bacterium clone F54*, *Uncultured bacterium clone ZBAF1-105*, *Uncultured bacterium clone SS-9*, and *Uncultured bacterium clone B13* in the AGS were identified as functional species in biodegradation of berberine and/or its metabolites. Both anaerobic and aerobic bacterial communities could adapt appropriately to different berberine selection pressures, since the functional species' identical functions ensured comparable pollutant removal performances. The information that has been provided in this study may help with future research in gaining a better understanding of the berberine biodegradation process.

**Supplementary Materials:** The following supporting information can be downloaded at: https://www.mdpi.com/article/10.3390/pr10122506/s1, Figure S1: The image of the hybrid ABR–AGS system; Figure S2: The change in the berberine concentration from A1 to A4; Table S1: Physical and chemical properties of inoculum; Table S2: The operation conditions of the hybrid ABR–AGS system; Table S3: The composition of influent wastewater.

**Author Contributions:** Conceptualization, P.Z.; methodology, J.L. and F.L.; investigation, Y.W. and R.M.; data curation, Y.W.; writing—original draft preparation, P.Z. and Y.W.; writing—review and editing, Y.L. and C.A.N.; project administration, P.Z.; funding acquisition, P.Z. All authors have read and agreed to the published version of the manuscript.

**Funding:** This research was funded by the National Key Research and Development Program of China (No. 2022YFC3203300), the Chinese Research Academy of Environmental Sciences Central Public Welfare Scientific Research Project (2022YSKY-63), and the National Major Scientific and Technological Projects for Water Pollution Control and Management (2017ZX07402003).

**Institutional Review Board Statement:** Not applicable.

**Informed Consent Statement:** Not applicable.

**Conflicts of Interest:** The authors declare no conflict of interest.

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
