# Peer review of "The Dynamic Shift of Bacterial Communities in Hybrid Anaerobic Baffled Reactor (ABR)—Aerobic Granules Process for Berberine Pharmaceutical Wastewater Treatment"

_processes, doi:10.3390/pr10122506_

Round 1

Reviewer 1 Report

This manuscript touches an important concern remediation concern of berberine from pharmaceutical wastewater. However, the quality of the manuscript could be improved if the comments I made below were to be considered and integrated. See below:

Please add line numbers to facilitate the review of the manuscript 

The following sentence has a typo: "The bacterial community dynamics was study using 16S rDNA clone library"

Section 1:

What concentration of the berberine is considered hazardous? Why was synthetic wastewater used in this study? Are there areas or industrial outlets where the berberine concentration in wastewater was high enough to become a source of concern? If so, please list them in this study.

Section 2.1:

Was the length/height ratio of the ABR randomly chosen? If not, what reference or logic was used for its design?

Are the authors referring to influent as opposed to influence in this sentence: "The angle of the baffle plate was 45°, which was beneficial for the influence entering the center of the upper flow chamber from the bottom, to make the sludge and influence can be mixed completely"?

Figure S1 in mentioned in section 2.1 but not displayed near the paragraph  that mentions it. 

Please label the figures individually for better legibility and flow of the manuscript.

"Furthermore, these findings indicated that a functional stable bacterial community for berberine degradation had already been established." The stability of an anaerobic bioreactor can be evaluated with the ratio between Volatile fatty acids and Alkalinity, why were these parameters not assessed to confirm the stability of the system?

Section 3.2, the Figures should be placed next to paragraphs where they are mentioned.

The acronyms OUT and OTU are used before being defined in the text.

Why were the bioreactors not operated longer? usually the stability and the scalability proneness of a system are evaluated with a system that remains highly performant and stable for the considerable operational time.

It could be more insightful to present some of the findings discussed in section 4 in a graph.

Reviewer 2 Report

In this manuscript, a lab-scale anaerobic baffled reactor-aerobic granular sludge process is utilized to treat berberine wastewater, and the dynamic shift of bacteria communities of the process is examined by 16S rDNA clone library. Few questions should be clarified further.

1. The inoculums for the startup of ABR reactor were obtained from the wastewater treatment facilities of a chemical synthetic pharmacy company. Consequently, after 25 days, the hybrid ABR-AGS system achieved high and stable berberine and COD removal rates. However, 25 days are quite short time for the stable of the anaerobic reactor. Does the chemical synthetic pharmacy company treat the wastewater containing berberine? If this is the truth, what effects will this brings to the results obtained?

2. Please provide the long-term performances of the process.

3. Page 15.

The authors declare that: For the AGS operation, the aerobic granules appeared on day 15, dominated on day 40, matured on day 80. However, the effluent quality of the hybrid ABR-AGS system achieved high and stable berberine and COD removal rates after 25 days. How to explain the differences between the two times? Does this mean the microbial community shift doesn’t affect the performance of the process?

4. P15

The authors declare that, berberine removal from synthetic wastewater was proposed using a hybrid ABR-AGS process. What is the novelty or significance of this paper? The authors should underline this problem in Introduction.
